# CoCDaR and mCoCDaR: New Approach for Measurement of Systemic Risk Contributions

**Rui Ding \*** and **Stan Uryasev**

Department of Applied Mathematics and Statistics, Stony Brook University, Stony Brook, NY 11794, USA; stanislav.uryasev@stonybrook.edu
**\*** Correspondence: rui.ding.1@stonybrook.edu

**Abstract:** Systemic risk is the risk that the distress of one or more institutions trigger a collapse of the entire financial system. We extend CoVaR (value-at-risk conditioned on an institution) and CoCVaR (conditional value-at-risk conditioned on an institution) systemic risk contribution measures and propose a new CoCDaR (conditional drawdown-at-risk conditioned on an institution) measure based on drawdowns. This new measure accounts for consecutive negative returns of a security, while CoVaR and CoCVaR combine together negative returns from different time periods. For instance, ten 2% consecutive losses resulting in 20% drawdown will be noticed by CoCDaR, while CoVaR and CoCVaR are not sensitive to relatively small one period losses. The proposed measure provides insights for systemic risks under extreme stresses related to drawdowns. CoCDaR and its multivariate version, mCoCDaR, estimate an impact on big cumulative losses of the entire financial system caused by an individual firm's distress. It can be used for ranking individual systemic risk contributions of financial institutions (banks). CoCDaR and mCoCDaR are computed with CVaR regression of drawdowns. Moreover, mCoCDaR can be used to estimate drawdowns of a security as a function of some other factors. For instance, we show how to perform fund drawdown style classification depending on drawdowns of indices. Case study results, data, and codes are posted on the web.

**Keywords:** systemic risk; conditional value-at-risk; CVaR; CVaR regression; drawdown; conditional drawdown-at-risk; fund style classification

---

## 1. Introduction

Systemic risk is the risk that the distress of one or more institutions triggers a collapse of the entire financial system. The CoVaR measure for systemic risk contributions was first proposed by Adrian and Brunnermeier (2008). This measure is the value-at-risk (VaR) of the financial system conditional on an institution (bank) being in financial distress. The systemic risk contribution of an institution is defined as a difference of VaR conditioning on the institution being under distress and being in its normal state. Huang and Uryasev (2017) replaced VaR by conditional value-at-risk (CVaR) and proposed the CoCVaR measure. CVaR has superior mathematical properties as compared to VaR; see, for instance, Rockafellar and Uryasev (2002). CVaR takes into account losses in the distribution tail, while VaR is not sensitive to outcomes in the tail.

Similar to Huang and Uryasev (2017), this paper is based on CVaR, but returns are replaced by drawdowns. The relevant risk measure is called conditional drawdown-at-risk (CDaR). By applying the CoCVaR approach to drawdowns, we defined CoCDaR. Therefore, CoCDaR is CDaR of the financial system conditioned on an institution in distress measured by drawdown. The intuition behind CDaR instead of VaR or CVaR is that these two measures do not take into account consecutive losses. As a result, small consecutive losses resulting in a large cumulative loss are not picked up

by VaR or CVaR. Drawdown, which is capturing cumulative losses, is popular in active portfolio management. The idea behind CoCDaR is that large drawdowns of financial institutions have a strong effect on the system as a whole. Hence, by conditioning on large drawdowns of institutions we can analyze systemic risk contributions (compared to effect of one-period negative returns).

We further extended CoCDaR with multiple regression framework and developed so-called mCoCDaR. This measure allows for multiple institutions being in distress, while CoVaR, CoCVaR and CoCDaR assume that only one institution is in distress and others are in normal states. Similar to mCoCDaR, we considered a multiple regression version of CoCVaR, called mCoCVaR. Therefore, mCoCVaR and mCoCDaR account for multiple marginal risk contributions of institutions and are well-defined Shapley values. This approach is motivated by the idea of identifying a risk contribution of each institution that is independent of contributions of other institutions. The estimation of CoCDaR and mCoCDaR was performed with CVaR regression developed in Rockafellar et al. (2014) and Golodnikov et al. (2019). The CVaR regression in CoCDaR uses drawdowns, while CoCVaR uses returns.

The mCoCDaR framework was also illustrated with fund style classification by using drawdowns instead of returns. This approach extends Bassett and Chen (2001), which used quantile regressions of fund returns depending on returns of indices. In addition, we have considered portfolio optimization formulations with CoCVaR and CoCDaR objectives and risk constraints.

CoCDaR and mCoCDaR approaches were demonstrated with a case study for the 10 largest USA banks. Furthermore, we have performed drawdown style classification of the Magellan fund using four stock indices. CVaR regression was implemented with Portfolio Safeguard (PSG) developed by AORDA (http://aorda.com). Case studies results and codes are posted on the web for verification purposes.

## 2. Methodology

### 2.1. Drawdown Definition

Suppose $r_1, \ldots, r_T$ are the rates of return of a risky instrument coming from a distribution of return random variable $X$. Let $\xi_t$ be the cumulative rate of return of the instrument for time $t$, which can be either uncompounded and defined by $\xi_t = \sum_{k=1}^{t} r_k$ or compounded and defined by $\xi_t = \prod_{k=1}^{t}(1 + r_k) - 1$. Further analysis in this section holds for either definition of the cumulative return, however, for the sake of tractability of optimization problems, $\xi_t$ is defined as uncompounded cumulative rate of return.

The drawdown of the instrument at time $t$ with $\tau$-window is defined as follows (see Chekhlov et al. (2005); Zabarankin et al. (2014)),

$$y_t = \max_{t_\tau \leq k \leq t} \xi_k - \xi_t, \qquad t_\tau = \max\{t - \tau, 1\}, \qquad t = 1, \ldots, T, \quad \tau = 1, \ldots, T. \tag{1}$$

At time $t$, the drawdown is the loss of the instrument, since a peak of $\xi_t$ that occurs within the $\tau$-window $[t_\tau, t]$ ($t_\tau = 1$ for $t \leq \tau$ and $t_\tau = t - \tau$ for $t > \tau$). If at time $t$, the cumulative rate of return $\xi_t$ is the highest on $[t_\tau, t]$, then $y_t = 0$. The drawdown is always nonnegative and is often referred to as underwater curve. It is zero for all time moments only if returns are nonnegative for all period. See Figure 1 for the illustration of the drawdown definition (the figure is borrowed from Zabarankin et al. (2014)).

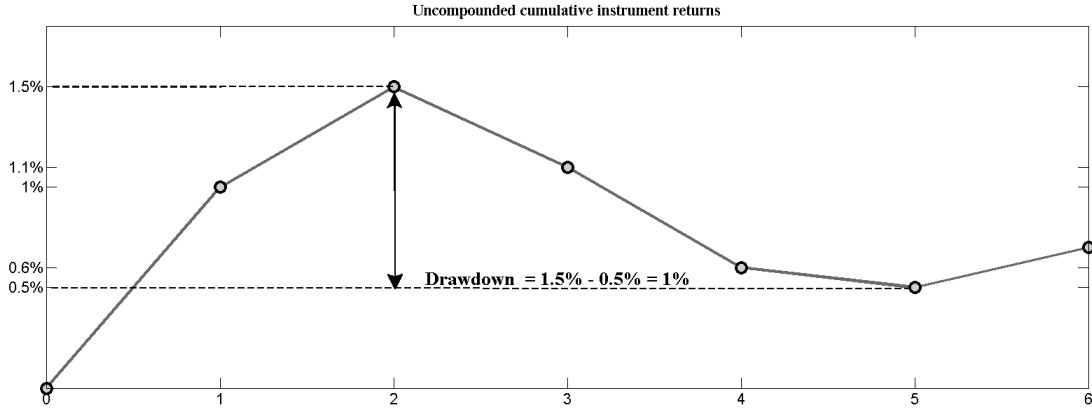

**Figure 1.** Drawdown example: the solid line is the uncompounded cumulative rate of return, which at time $t$ is the sum of rates of return over periods $1, \ldots, t$. Here, $\tau = 6$. For $t = 5$, $\xi_5 = 0.5\%$, whereas the maximum of $\xi_t$ over time moments preceding $t = 5$ occurs at $t = 2$ with $\xi_2 = 1.5\%$. Consequently, $y_5 = 1.5\% - 0.5\% = 1\%$. The instrument maximum drawdown over time period $[0, 6]$ occurs at $t = 5$.

*2.2. CoCDaR Definitions*

Conditional value-at-risk (CVaR) of a random variable $X$ (see Rockafellar and Uryasev (2000, 2002)), can be defined as follows,

$$CVaR_\alpha(X) = \min_C \left\{ C + \frac{1}{1 - \alpha} E[(X - C)^+] \right\},$$

where $A^+ = \max\{0, A\}$. The $\alpha$-conditional drawdown is an expectation over the worst $1 - \alpha$ drawdowns occurring in the considered horizon. We can look at $\{y_t\}_{1 \leq t \leq T}$ as a nonlinear transformation of observations from the random return variable $X$ and denote the random variable for drawdowns by $Y$. The conditional drawdown-at-risk (CDaR) for $X$ is defined as CVaR of $Y$:

$$CDaR_\alpha(X) = CVaR_\alpha(Y).$$

Let $X^{sys}$ denote return of a financial system and let returns of financial institutions $i = 1, \ldots, I$ be denoted as $X^i$. Given a sample path of data $\{x_t^{sys}, x_t^1, \ldots, x_t^I\}_{1 \leq t \leq T}$, we can obtain the drawdown observations for the financial system as well as all the institutions and denote them by $\{y_t^{sys}, d_t^1, \ldots, d_t^I\}_{1 \leq t \leq T}$. Let $Y^{sys}, D^1, \ldots, D^I$ denote random variables associated with these observations. Similar to Huang and Uryasev (2017), we can define CoCDaR as:

$$CoCDaR_\alpha^{sys|i} = CDaR_\alpha(X^{sys}|X^i, M_1, \ldots, M_n) = CVaR_\alpha(Y^{sys}|D^i, M_1, \ldots, M_n).$$

Here $M_1, \ldots, M_n$ are state factor variables which we define in the next section. They are the lagged system variables used in Huang and Uryasev (2017), but transformed to provide more explanatory powers in drawdown regression (considered later on).

$CoCDaR_\alpha^{sys|i}$ gives a measure of CDaR of a system (index) conditioning on the drawdown level of an individual institution (stock) $i$, along with some other state variables. By using drawdown instead of return, we are looking specifically at the impact of individual institution drawdown to the entire financial system drawdown as a measure of systemic risk contribution (that takes into account consecutive distress periods). This intuition will be further developed in Section 2.6.

*2.3. State Variables*

Lagged state variables $M_{1,t-1}, \ldots, M_{n,t-1}$ used in the regression in the following section were introduced by Adrian and Brunnermeier (2008):

(1) VIX = The Chicago Board Options Exchange Volatility Index;

(2) Liquidity Spread = A short-term liquidity spread, defined as the difference between the three-month repurchase agreement rate and the three-month treasury bill rate;

(3) 3M Treasury Change = The change in the three-month T-bill rate;

(4) Term Spread Change = The change in the slope of the yield curve, measured by the yield spread between the ten-year treasury rate and the three-month bill rate;

(5) Credit Spread Change = The change in the credit spread between Baa-rated bonds and the treasury rate;

(6) Equity Returns = The equity market return from S&P 500 Index;

(7) Real Estate Excess Return = The real estate sector return in excess of the market return.

*2.4. Estimation of CoCDaR*

Consider the following regression similar to Adrian and Brunnermeier (2008) and Huang and Uryasev (2017),

$$Y_t^{sys} \sim \beta_0 + \beta_1 D_t^i + \omega_1 M_{t-1,1} + ... + \omega_n M_{t-1,n} .$$

We define the residual random variable as:

$$L = Y^{sys} - (\beta_0 + \beta_1 D^i + \omega_1 M_1 + ... + \omega_n M_n) .$$

This regression problem uses a single institution's drawdown and lagged state variables as factors to model the drawdown of the financial system. We have $T$ observations of the system drawdowns, drawdowns of institution $i$, and the state factors. We next perform a CVaR regression with the above model and find CVaR of the system's drawdown conditioned on drawdowns of institution $i$. Here, the state factors are cumulative changes of each fundamental factor in the period of the current drawdown of the financial system. In particular, VIX and liquidity spread are given in numbers so we calculate the time lagged difference of them in the current period of system drawdown. The other state factors are given in percentage changes, so we calculate their cumulative changes in the current period of system drawdown.

For each time step $t$, we consider the system cumulative returns $\xi_t^{sys}$ and find the historic peak time (used in drawdown definition), denoted by

$$\nu(t) = \arg\max_{t_\tau \leq s \leq t} \xi_s^{sys} . \tag{2}$$

Let the original state variable values (numeric or cumulative changes in percentage) be denoted by $m_t$. The transformed state variables for the CoCDaR regression are hence defined for each $j = 1, \ldots, n$:

$$M_{t-1,j} = m_{t-1,j} - m_{\nu(t),j} .$$

The estimate of the $\alpha$-CVaR of $Y^{sys}$ can be obtained by minimizing the CVaR (superquantile) error from Rockafellar et al. (2014):

$$\mathcal{E}_\alpha^{CVaR}(L) = \frac{1}{1-\alpha} \int_0^1 CVaR_\gamma^+(L) \, d\gamma - E[L] . \tag{3}$$

Golodnikov et al. (2019) proved that minimization of error (3) for CVaR regression can be reduced to the minimization of the Rockafellar error (convex and liner programming formulations are in Appendix A, Golodnikov et al. (2019)). The Rockafellar error belongs to the mixed quantile quadrangle, as defined by Rockafellar and Uryasev (2013). For given confidence levels $\alpha_k \in (0,1)$ and weights $\lambda_k > 0$, $k = 1, \ldots, K$ such that $\sum_{k=1}^K \lambda_k = 1$, the Rockafellar error equals:

$$\mathcal{E}^{ROC}(L) = \min_{C_1,\dots,C_K} \left\{ \sum_{k=1}^{K} \lambda_k \mathcal{E}_{\alpha_k}^{KB}(L - C_k) \mid \sum_{k=1}^{K} \lambda_k C_k = 0 \right\} , \tag{4}$$

where the rescaled Koenker–Bassett (KB) error equals:

$$\mathcal{E}_{\alpha}^{KB}(L) = E\left[ \frac{\alpha}{1-\alpha} L^+ + (-L)^+ \right] . \tag{5}$$

Koenker and Bassett (1978) suggested estimating a conditional quantile by minimizing error (5). Since CVaR is an integral of quantile (VaR), then it is not surprising that CVaR can be estimated with Rockafellar error (4) which is a weighted average of KB-errors. The Rockafellar error is quite a complicated function: it is a minimum of a convex nonsmooth function with respect to variables $C_1, \dots, C_K$ with a linear constraint. However, since this error is a convex piece-wise linear function, it can be minimized very efficiently; see for instance results of numerical experiments in Golodnikov et al. (2019). The resulting coefficients will provide an estimate of the $\alpha$-CVaR of the dependent variable conditioned on the independent variables.

Denote by $\hat{\beta}_0^{\alpha}, \hat{\beta}_1^{\alpha}, \hat{\omega}_1^{\alpha}, \dots, \hat{\omega}_n^{\alpha}$ the regression coefficients obtained by minimizing the Rockafellar error (4). CoCVaR of the system's drawdown, which is CoCDaR of the system, is estimated by:

$$CoCDaR_{t,\alpha}^{sys} = \hat{\beta}_0^{\alpha} + \hat{\beta}_1^{\alpha} D_t^i + \hat{\omega}_1^{\alpha} M_{t-1,1} + \dots + \hat{\omega}_n^{\alpha} M_{t-1,n} .$$

This regression estimation is done for every institution, $i = 1, \dots, I$.

*2.5. Institutional Drawdown-at-Risk*

To calculate system CoCDaR at some risk level conditioned on institution $i$ being in drawdown distress, we need to set an institutional distress level $D_t^i$.

$\alpha$-value-at-risk (VaR), which is also $\alpha$-quantile, of a random loss variable $L$ is defined as:

$$VaR_{\alpha}(L) = \inf\{x : F_L(x) \geq \alpha\} .$$

We define $\alpha$-drawdown-at-risk ($\alpha$-DaR) of an institution $i$ as the $\alpha$-quantile (VaR) of the drawdown loss random variable $D^i$ corresponding to its return random variable $X^i$, where $\alpha \in [0,1]$,

$$DaR_{\alpha}(X^i) = VaR_{\alpha}(D^i) .$$

Similar to Huang and Uryasev (2017), we can use quantile regression for estimation of $\alpha$-DaR:

$$D_t^i \sim \gamma_0^i + \gamma_1^i M_{t-1,1}^i + \dots + \gamma_n^i M_{t-1,n}^i .$$

Here, the state factors $M_1^i, \dots, M_n^i$ are defined differently compared to the CoCDaR regression. They are the same fundamental factor changes but calculated in the current period of each institution drawdown. Define $v_i(t) = argmax_{t_\tau \leq s \leq t}\, \xi_s^i$ for each institution $i = 1, \dots, I$, where $\xi_t^i$ are the cumulative returns observations. The transformed state variables for the DaR regression are hence defined for each $j = 1, \dots, n$:

$$M_{t-1,j}^i = m_{t-1,j} - m_{v_i(t),j} .$$

Let the residual term be denoted as:

$$G^i = D^i - (\gamma_0^i + \gamma_1^i M_1^i + \dots + \gamma_n^i M_n^i) .$$

By minimizing KB-error, $\mathcal{E}_{\alpha}^{KB}(G^i)$, we find coefficients $\hat{\gamma}_0^i, \dots, \hat{\gamma}_n^i$ and estimate the $\alpha$-quantile of $D_t^i$:

$$DaR_{t,\alpha}^i = \hat{\gamma}_0^i + \hat{\gamma}_1^i M_{t-1,1}^i + \dots + \hat{\gamma}_n^i M_{t-1,n}^i .$$

### 2.6. Sytemic Risk Contribution

We have defined the CoCDaR measure and suggested an estimation procedure with CVaR regression. Next we show how to use this measure for systemic risk contribution measurement. We follow definitions from Huang and Uryasev (2017) and define:

$$X_t^{sys} = 100 \ln \frac{I_t}{I_{t-1}} ,$$

as the system's return variable which is the log return of the index value, such as the Dow Jones Index. Similarly, the *i*-th financial institution log return $X_t^i$ is defined as:

$$X_t^i = 100 \ln \frac{P_t^i}{P_{t-1}^i} ,$$

where $P_t^i$ is the closing price of institution *i* at time *t*.

Using the definitions in previous sections, we get the drawdown observations $Y_t^{sys}, D_t^i$ for the financial system and an institution *i*. We also have state factors, $M_{t-1,1}, ..., M_{t-1,n}$, for every time moment *t* is in the considered horizon.

We first perform the quantile regression in Section 2.5 to estimate $DaR_{t,\alpha'}^i$ for all *t* for two particular levels: $\alpha_1' = 0.9$ and $\alpha_2' = 0.5$. The level $\alpha_1' = 0.9$ corresponds to the distress level of the institution in terms of its drawdown and $\alpha_2' = 0.5$ corresponds to the median (normal) state of the institution.

Next we perform the CVaR regression from Section 2.4 and obtain an estimate of the $\alpha$-CoCDaR of the financial system conditioned on the drawdown level of institution *i* and state factors. Here $\alpha$ is different from $\alpha'$ used in the previous quantile regression. For every time step *t*, we calculate:

$$CoCDaR_{t,\alpha}^{sys|D_t^i=DaR_{t,\alpha'}^i} = \hat{\beta}_0^\alpha + \hat{\beta}_1^\alpha DaR_{t,\alpha'}^i + \hat{\omega}_1^\alpha M_{t-1,1} + ... + \hat{\omega}_n^\alpha M_{t-1,n} .$$

By choosing $\alpha_1' = 0.9$ and $\alpha_2' = 0.5$ for the DaR level for an individual institution and selecting a separate risk level $\alpha$ for system CoCDaR, we obtain:

$$\Delta CoCDaR_{t,\alpha}^{sys|i} = CoCDaR_{t,\alpha}^{sys|D_t^i=DaR_{t,0.9}^i} - CoCDaR_{t,\alpha}^{sys|D_t^i=DaR_{t,0.5}^i} .$$

This difference is defined as the systemic drawdown risk contribution of institution *i* to the financial system at the selected risk level $\alpha$. More concretely, it calculates the difference in conditional drawdown-at-risk values of the financial system given that the drawdown level of institution *i* is at its distress level or its normal level as a measure of systemic drawdown risk contribution.

### 2.7. mCoCDaR Definition

Using the same set of state factors and extending the idea of CoCDaR as a measure of systemic risk contribution, we propose a more comprehensive measure called multiple-CoCDaR, which measures the conditional drawdown-at-risk of the financial system conditioned on the distress levels of all *I* institutions being considered. The idea is an extension of the CoCDaR approach defined above by combining it with a generalization of the multiple-CoVaR method defined in Bernardi et al. (2013) and Bernardi and Petrella (2014). In their paper, a similar approach was developed that defines conditional tail risk of a system/institution conditioned on the distress level of multiple institutions at the same time. A similar approach was also seen in Cao (2013). Different from their methods, our approach uses a simple multiple regression formulation. In the multiple regression framework, we can measure risk contribution of an institution by taking the difference between CoCDaR values of the system under different drawdown levels of that institution alone, while holding other institutions' drawdown values fixed at their normal levels. We define mCoCDaR as:

$$mCoCDaR_{\alpha}^{sys|1,...,I} = CDaR_{\alpha}(X^{sys}|X^1,...,X^I,M_1,...,M_n) = CVaR_{\alpha}(Y^{sys}|D^1,...,D^I,M_1,...,M_n) \,.$$

### 2.8. Estimation of mCoCDaR

Consider the following regression using the same set of state factors as in CoCDaR regression,

$$Y_t^{sys} \sim \beta_0 + \beta_1 D_t^1 + ... + \beta_I D_t^I + \omega_1 M_{t-1,1} + ... + \omega_n M_{t-1,n} \,,$$

with residual:

$$L = Y^{sys} - (\beta_0 + \beta_1 D^1 + ... + \beta_I D^I + \omega_1 M_1 + ... + \omega_n M_n) \,.$$

This regression problem uses $I$ institutions drawdowns and lagged state variables as factors to model the drawdown of the financial system. We have $T$ observations for the system drawdown random variable, the $I$ institutions' drawdown random variables, and the state factors' random variables. We next perform a CVaR regression of the above model to find CVaR of the system drawdown conditioned on all $I$ institution drawdown. Denote by $\hat{\beta}_0^{\alpha}, \hat{\beta}_1^{\alpha}, ..., \hat{\beta}_I^{\alpha}, \hat{\omega}_1^{\alpha}, ..., \hat{\omega}_n^{\alpha}$ coefficients obtained by minimizing Rockafellar error (4). These coefficients allow one to compute the $\alpha$-CDaR of the financial system conditioned on drawdowns of all the institutions and state factors.

The multiple-CoCVaR of the system drawdown, which is equivalent to the multiple-CoCDaR of the financial system, is estimated by:

$$mCoCDaR_{t,\alpha}^{sys} = \hat{\beta}_0^{\alpha} + \hat{\beta}_1^{\alpha} D_t^1 + ... + \hat{\beta}_I^{\alpha} D_t^I + \hat{\omega}_1^{\alpha} M_{t-1,1} + ... + \hat{\omega}_n^{\alpha} M_{t-1,n} \,.$$

This procedure applies one regression problem using all institutions' drawdown observations to obtain coefficient estimates. The institutional DaRs are calculated exactly the same way as in Section 2.5.

### 2.9. Sytemic Risk Contribution using mCoCDaR

We have defined mCoCDaR measure and the estimation procedure with CVaR regression. We use this measure for systemic risk contribution measurement, following the definitions in Section 2.4. The drawdown observations are denoted by $Y_t^{sys}, D_t^1, ..., D_t^I$ for the financial system and all $I$ institutions respectively. We also have lagged state variables $M_{t-1,1}, ..., M_{t-1,n}$ for every time moment $t$.

We first perform the quantile regression in Section 2.5 to estimate $DaR_{t,\alpha'}^i$ for all $t$ and for all $i$ for two particular levels: $\alpha'_{i,1} = 0.9$ and $\alpha'_{i,2} = 0.5$. Level $\alpha'_{i,1} = 0.9$ corresponds to the distress level of the $i$-th institution in terms of its drawdowns and $\alpha'_{i,2} = 0.5$ corresponds to its median (normal).

Next we perform the CVaR regression from Section 2.8 and estimate the financial system's conditional drawdown-at-risk conditioned on the drawdown levels of all $I$ institutions and state factors. For every time step $t$, we calculate,

$$mCoCDaR_{t,\alpha}^{sys|D_t^1=DaR_{t,\alpha'_1}^1,...,D_t^I=DaR_{t,\alpha'_I}^I} = \hat{\beta}_0^{\alpha} + \hat{\beta}_1^{\alpha} DaR_{t,\alpha'_1}^1 + ... + \hat{\beta}_I^{\alpha} DaR_{t,\alpha'_I}^I + \hat{\omega}_1^{\alpha} M_{t-1,1} + ... + \hat{\omega}_n^{\alpha} M_{t-1,n} \,.$$

Now, to analyze the effect of a single institution $i$ on the financial system, we compute the mCoCDaR values based on $\alpha'_{i,1} = 0.9$ and $\alpha'_{i,2} = 0.5$, while holding $\alpha'_{-i} = 0.5$ fixed where $-i$ means all the institutions other than $i$, and calculate the difference in mCoCDaR,

$$\Delta mCoCDaR_{t,\alpha}^{sys|i} = mCoCDaR_{t,\alpha}^{sys|D_t^i=DaR_{t,0.9}^i,D_t^{-i}=DaR_{t,0.5}^{-i}} - mCoCDaR_{t,\alpha}^{sys|D_t^i=DaR_{t,0.5}^i,D_t^{-i}=DaR_{t,0.5}^{-i}} \,.$$

This difference is the incremental/marginal systemic drawdown risk contribution of the distress of institution $i$ to the financial system, while other institutions are at their normal states.

We can switch back to the original return observations instead of the drawdown observations and perform the regression procedure in Sections 2.7 and 2.8. This way we get another measure for systemic risk contribution which we call mCoCVaR. It measures the incremental/marginal conditional

value-at-risk of the financial system's returns conditioned on one institution's return being in distress while all the other institutions are in their normal states.

## 2.10. Advantages of mCoCDaR and mCoCVaR

As we have seen in the previous section, the multiple version of the systemic risk conditional estimation provides a more general framework to analyze the effect on the financial system posed by a particular institution's distress, or perhaps multiple financial institutions' joint distress. It is based on the idea that during periods of financial instability, several institutions may experience financial distress at the same time, so their risk contributions can be highly correlated. Switching from the CoCVaR and CoCDaR to their multiple regression versions helps to mitigate these dependencies on risk contribution measures.

With mCoCDaR, we can measure the contribution to the financial system's conditional drawdown-at-risk conditioned on the drawdown levels of two institutions $i, j$ as follows,

$$\Delta mCoCDaR_{t,\alpha}^{sys|i,j} = mCoCDaR_{t,\alpha}^{sys|D_t^i=DaR_{t,0.9}^i,D_t^j=DaR_{t,0.9}^j,D_t^{-i,j}=DaR_{t,0.5}^{-i,j}}$$

$$-mCoCDaR_{t,\alpha}^{sys|D_t^i=DaR_{t,0.5}^i,D_t^j=DaR_{t,0.5}^j,D_t^{-i,j}=DaR_{t,0.5}^{-i,j}} .$$

There is a lot of flexibility on the risk levels to choose for this type of analysis, which means the DaR level for the two institutions in distress can be set differently. This approach considers the joint impact of two institutions without distinguishing their respective contributions, which is not included in the original framework without using multiple regression. The flexibility given by mCoCDaR, and similarly mCoCVaR, does not come at additional computation costs. In fact, by combining all institutions in one regression problem, we save computational time.

Another advantage of the multiple-CoCVaR and multiple-CoCDaR are their consistency as risk distribution measures. Bernardi et al. (2013) noticed that the original $\Delta CoVaR^{sys|i}$ is not a desirable risk distribution measure, because summing up $\Delta CoVaR^{sys|i}$ for all institutions $i$ does not generally equal their overall effect on the system. This issue is addressed in Bernardi et al. (2013) and Bernardi and Petrella (2014) via the Shapley value, which transforms the calculated contribution using $\Delta Multiple - CoVaR$ to a Shapley value for each institution so that their contribution adds up to the joint contribution of all institutions together on the system. The Shapley value methodology was originally proposed to measure shared utility or cost among participants of a cooperative game.

We observe that the individual risk contribution calculated with $\Delta mCoCVaR$ or $\Delta mCoCDaR$ does not have this drawback. For instance, for mCoCDaR:

$$mCoCDaR_{t,\alpha}^{sys|D_t^1=DaR_{t,\alpha_1'}^1,...,D_t^I=DaR_{t,\alpha_I'}^I} = \hat{\beta}_0^\alpha + \hat{\beta}_1^\alpha DaR_{t,\alpha_1'}^1 + ... + \hat{\beta}_I^\alpha DaR_{t,\alpha_I'}^I + \hat{\omega}_1^\alpha M_{t-1,1} + ... + \hat{\omega}_n^\alpha M_{t-1,n} .$$

Once we have estimated the coefficients via CVaR regression, we can calculate the individual contribution of institution $i$ entering stress level 0.9 as:

$$\Delta mCoCDaR_{t,\alpha}^{sys|i} = mCoCDaR_{t,\alpha}^{sys|D_t^i=DaR_{t,0.9}^i,D_t^{-i}=DaR_{t,0.5}^{-i}} - mCoCDaR_{t,\alpha}^{sys|D_t^i=DaR_{t,0.5}^i,D_t^{-i}=DaR_{t,0.5}^{-i}}$$

$$= \hat{\beta}_i^\alpha(DaR_{t,0.9}^i - DaR_{t,0.5}^i) \equiv V_{sys}(i) .$$

The total contribution of all financial institutions distress on the systemic risk is:

$$\Delta mCoCDaR_{t,\alpha}^{sys|1,...,I}$$

$$= mCoCDaR_{t,\alpha}^{sys|D_t^1 = DaR_{t,0.9}^1,...,D_t^I=DaR_{t,0.9}^I} - mCoCDaR_{t,\alpha}^{sys|D_t^1=DaR_{t,0.5}^1,...,D_t^I=DaR_{t,0.5}^I}$$

$$= \sum_{i=1}^{I} \hat{\beta}_i^{\alpha} (DaR_{t,0.9}^i - DaR_{t,0.5}^i) = \sum_{i=1}^{I} V_{sys}(i) .$$

A similar statement is valid for *mCoCVaR*. The entire systemic risk is exactly distributed to its institutional components. $\Delta mCoCVaR$ and $\Delta mCoCDaR$ are both Shapley value functions, denoted by $V_{sys}(i)$ for contributor $i$, such that they satisfy the following desirable mathematical properties as outlined in Bernardi et al. (2013). Let $\mathcal{S}$ be a set of $I$ institutions:

(1) Efficiency: $\sum_{i=1}^{I} V_{sys}(i) = V_{sys}(\mathcal{S})$. This axiom states that the total risk is distributed to participants.

(2) Symmetry: For $i \neq j$ such that $V_{sys}(H \cup i) = V_{sys}(H \cup j), \forall H$ such that $i, j \notin H$, then $V_{sys}(i) = V_{sys}(j)$. This axiom states that the contribution measure is permutation invariant and fair for all contributors.

(3) Dummy axiom: $V_{sys}(H \cup i) = V_{sys}(i), \forall i \in H$ and $H \supseteq \mathcal{S}$. This means, if the risk of institution $j$ is independent of all other institutions, then its risk contribution to the system should be its own risk. Generally, in CoCVaR and CoCDaR approaches (also CoVaR), the risks are not orthogonal among institutions. Hence, their ranking should differ from those provided by a Shapley value measure such as *mCoCVaR* and *mCoCDaR*.

(4) Linearity (additivity): If $i, j \in H, i \neq j$ are two institutions, where $V_{sys}(i) \neq V_{sys}(j)$, let $w_i > 0, w_j > 0, k = w_i i + w_j j$, then new combined risk contributions equal the weighted average of individual risk contributions: $V_{sys}(k) = w_i V_{sys}(i) + w_j V_{sys}(j)$.

(5) Zero player: If $i \notin S, V_{sys}(i) = 0$. A null player receives zero risk contribution.

*2.11. mCoCDaR Versus mCoCVaR*

We do not claim that one of the considered risk measures is better for analyzing systemic risk contributions than the other one. CoCVaR is concerned with the conditional risk in terms of the returns' tail behavior, while CoCDaR is concerned with the conditional risk in terms of the drawdowns. These measures have a nonlinear relationship embedded in their definitions.

When a financial system's large drawdowns are significantly correlated with large drawdowns of some particular institutions, it can be hypothesized that the CoCDaR measure will provide a more reasonable estimate of the risk contributions and therefore give a more reasonable ranking of the systemic risk contribution of each institution. This can be generalized to comparing mCoCVaR and mCoCDaR measures which are proposed in this work. Another intuition for using drawdown based approaches is that drawdown measures a psychological effect from a consistent distress in stock returns.

## 3. Case Studies

This case study uses data from CoCVaR paper Huang and Uryasev (2017), which is posted at this link[1]. Codes, data, calculation results for this case study are posted at this link[2].

We first computed the drawdowns from the return data and transformed the state factors corresponding to each regression problem. Next, we proceeded to the quantile regressions on institutional drawdowns and the CVaR regression on the system's drawdowns. The CVaR regression is implemented using Portfolio Safeguard (PSG)[3] in MATLAB environment. PSG includes efficiently implemented (precoded) Koenker–Bassett, Rockafellar and CVaR errors.

---

[1]   http://uryasev.ams.stonybrook.edu/index.php/research/testproblems/financial_engineering/case-study-cocvar-approach-risk-contribution-measurement

[2]   http://uryasev.ams.stonybrook.edu/index.php/research/testproblems/financial_engineering/case-study-cocdar-approach-systemic-risk-contribution-measurement

[3]   Portfolio Safeguard (PSG) is a product of American Optimal Decisions: http://aorda.com

### 3.1. Financial Institutions

We consider the ten largest publicly traded banks in the USA as of 31 December 2014:

1.  JP Morgan Chase & Company (JPM)
2.  Bank of America (BAC)
3.  Citigroup Inc (C)
4.  Wells Fargo & Company (WFC)
5.  The Bank of New York Mellon Corporation (BK)
6.  US Bancorp (USB)
7.  Capital One Financial Corporation (COF)
8.  PNC Financial Services Group Inc (PNC)
9.  State Street Corporation (STT)
10. The BB&T Corporation (BBT)

Data period is from 18 February 2000 to 30 January 2015. Closing prices are downloaded from Yahoo Finance for the Dow Jones US Financial Index and the financial institutions.

### 3.2. CoCVaR Calculation Results

In addition to numerical results presented in the following sections for new measures, we calculated systemic risk contributions based on the CoCVaR method. We reproduced the case study described in Section 3.3.4 of Huang and Uryasev (2017) with corrected input data (corrected the wrong sign in return data of financial instruments). We considered negative returns (losses) for each bank and the index. We averaged each bank's contributions across time and ranked them accordingly, where larger values correspond to stronger contributions to system's CoCVaR (the units of all reported values are 100%):

| | |
|---|---|
| 1. WFC: 0.03608 | 6. STT: 0.02905 |
| 2. BBT: 0.03210 | 7. COF: 0.02776 |
| 3. PNC: 0.03089 | 8. BK : 0.02740 |
| 4. JPM: 0.03077 | 9. USB: 0.02341 |
| 5. BAC: 0.03063 | 10. C : 0.00187 |

### 3.3. CoCDaR Calculation Results

The drawdown-at-risk values of each institution at two different risk levels, $\alpha'_1 = 0.9$ for distress level and $\alpha'_2 = 0.5$ for normal level, are computed using the quantile regression defined in Section 2.5. The CoCDaR values of the system at a specific risk level $\alpha = 0.9$ conditioned on each institution's DaR being at a distress level and a normal level are respectively computed based on the CVaR regression in Section 2.4. Following Section 2.6, the difference in CoCDaR values is taken and this results in a time series of $\Delta CoCDaR_{t,\alpha}^{sys|i}$ for each institution $i$ and for each observation time $t$.

We observe that the quantile regression for DaR using the state variables as regressors yields different behaviors for different institutions. Responses are different for the state variables: some are positive while others are negative. This is true for both the distress level and the normal level. The pseudo $R^2$ metrics for these quantile regressions are generally between 0.5 to 0.7, which indicates a descent level of explanatory power as compared with using just the original state factors. The observation is consistent with that made in Huang and Uryasev (2017). The CoCDaR and DaR calculation results are posted in the CoCDaR case study[2], see Problems 1 and 3. For the ten CVaR regressions of index drawdowns on the state variables and respective institution drawdowns, we observe that the coefficients for each factor typically have the same sign (with a few exceptions). The pseudo $R^2$ for CoCDaR regressions are all above 0.8.

We averaged each bank's contributions to CoCDaR across time and ranked the ten banks accordingly, where larger values correspond to stronger contributions to system's CoCDaR:

1. WFC: 0.27695 [1]
2. BAC: 0.22285 [5]
3. BBT: 0.07073 [2]
4. COF: 0.06107 [7]
5. USB: 0.05502 [9]

6. BK: 0.03564 [8]
7. PNC: 0.02558 [3]
8. JPM: 0.02306 [4]
9. STT: 0.00242 [6]
10. C: –0.01390 [10]

The number in brackets is the ranking based on $\Delta CoCVaR$ in Section 3.2. Results show that only Citigroup Inc. has negative CoCDaR contribution to the index on average, hinting that its drawdowns could have a negative correlation with index drawdowns. All other institutions are contributing positively to the system's conditional drawdown-at-risk.

In particular, PNC Financial Services Group Inc (PNC) was ranked third by $\Delta CoCVaR$ but ranked seventh by $\Delta CoCDaR$. On the other hand, Bank of America (BAC) was ranked fifth by $\Delta CoCVaR$ but ranked second by $\Delta CoCDaR$. Clearly, these two approaches provide different perspectives.

*3.4. mCoCVaR Calculation Results*

This section demonstrates the performance of suggested mCoCVaR, which is the multiple version of the CoCVaR approach developed in Huang and Uryasev (2017). We begin by performing the mCoCVaR analysis of the ten financial institutions in one CVaR regression. The pseudo $R^2$ for mCoCVaR regression is 0.76. The value-at-risk for normal and distress states are calculated for every institution respectively using quantile regressions on the original state variables. The procedure for VaR calculation is described in Huang and Uryasev (2017), Sections 2.3 and 3.3.2. By holding all other institutions' return values to their VaR values in a normal state (which corresponds to the median) and looking at the differences resulting from changing one particular institution's return value to its VaR value in a distress state, we obtain a time series of $\Delta mCoCVaR_{t,\alpha}^{sys|i}$ for each institution and for each observation time $t$. We averaged each bank's contributions to mCoCVaR across time and ranked the ten banks accordingly, where larger values correspond to stronger contributions to system's mCoCVaR:

1. BBT: 0.00813 [2]
2. BAC: 0.00721 [5]
3. BK: 0.00619 [8]
4. JPM: 0.00598 [4]
5. PNC: 0.00494 [3]

6. STT: 0.00463 [8]
7. WFC: 0.00306 [1]
8. COF: 0.00266 [7]
9. USB: 0.00149 [9]
10. C: 0.00062 [10]

The number in the bracket is the ranking according to $\Delta CoCVaR$ in Section 3.2. The results based on $\Delta mCoCVaR$ are similar to those based on $\Delta CoCVaR$, but there are some significant differences. For instance, WFC, originally ranked the highest, dropped to the seventh place in this new ranking. This might have been caused by its returns having a high correlation to returns of other institutions, for example The BB&T Corporation (BBT). This effect is neglected in the previous CoCVaR method, but in our multiple regression setting, by explicitly fixing the other institutions' returns to their respective normal states, we are analyzing the marginal impact of WFC's distress. Hence, the drop in ranking may indicate that WFC is not a key systemic risk contributor in the sense that its risk contributions are dependent on the high risk contributions of other institutions. Clearly, CoCVaR and mCoCVaR provide different perspectives regarding the ranking of financial institutions' risk contributions.

*3.5. mCoCDaR Results*

The drawdown-at-risk of each institution at two different risk levels, $\alpha'_1 = 0.9$ for distress level and $\alpha'_2 = 0.5$ for normal level, are computed using the quantile regression defined in Section 2.5; this step is identical to the first step performed in Section 3.3. The mCoCDaR values of the system at a specific risk level $\alpha = 0.9$ conditioned on each institution's DaR being at a distress level and a normal level are computed respectively based on the CVaR regression with multiple institutions as specified in Section 2.8. Following Section 2.9, the difference in mCoCDaR values is taken and this results in a time series of $\Delta mCoCDaR^{sys|i}_{t,\alpha}$ for each institution $i$ and for each observation time $t$.

Since we are using the same quantile estimates for DaR, we obtained the same observations as that in Section 3.3. For the CVaR regression of the drawdowns of the index on the state variables and the institution drawdowns, we observe that some institutions' regression coefficients are positive in the CVaR regression, while others are negative. The pseudo $R^2$ for mCoCDaR regression is 0.9. The mCoCDaR and DaR results are posted in the CoCDaR case study[2], see Problems 2 and 3.

We averaged each bank's contributions to mCoCDaR across time and ranked the ten banks accordingly, where larger values correspond to stronger contributions to system's mCoCDaR:

1. BAC: 0.20572 [2]
2. BBT: 0.02964 [3]
3. USB: 0.02485 [5]
4. STT: 0.02011 [9]
5. COF: 0.01749 [4]
6. BK: 0.01548 [6]
7. C: 0.00434 [10]
8. PNC: –0.01353 [7]
9. JPM: –0.01353 [8]
10. WFC: –0.06106 [1]

The number in the bracket is the ranking according to $\Delta CoCDaR$. $\Delta mCoCDaR$ and $\Delta CoCDaR$ rankings are mostly similar, yet have some interesting differences as well. While WFC is ranked highest by $\Delta CoCDaR$, it is ranked last by $\Delta mCoCDaR$. This observation coincides with what we saw in Section 3.4, indicating the high correlation that WFC might have with other top risk contributors such as BB&T and BAC. Furthermore, while STT is ranked second last by $\Delta CoCDaR$, it is ranked fourth by $\Delta mCoCDaR$.

*3.6. Comparative Summary of the Proposed Methods*

Table 1 provides a complete summary of the rankings of the ten banks with the four risk measures.

Compared with CoCVaR, CoCDaR takes into account drawdowns and focuses on consecutive losses. Using drawdowns is particularly insightful because drawdowns identify cumulative losses (negative cumulative returns), hence the dependencies between institutions and the system in "good" times are ignored. Dependencies in "bad" times are captured, which is important for risk analysis. We observe that CoCVaR and CoCDaR may provide very different rankings. For instance, USB with mCoCDaR and CoCDaR are ranked 3 and 5, accordingly (i.e., BAC is a top contributor), but with mCoCVaR and CoCVaR it is ranked 9 (i.e., close to bottom contributor). Even more surprisingly, JPM is ranked 9 and 8 with mCoCDaR and CoCDaR, but ranked 4 with mCoCVaR and CoCVaR.

mCoCVaR and mCoCDaR approaches add further insights to CoCVaR and CoCDaR, since they employ a multiple regression that marginalizes the systemic risk contributions of individual institutions. Running the multiple regression instead of individual ones enables us to look at institutions' contributions in a unified way, since their fraction contributions sum up to one.

Risk contributions based on CoVaR and CoCVaR measures, as a function of time, demonstrate a similar pattern for different institutions, see Huang and Uryasev (2017). This is probably because the methodology is based on separate regression for each institution. On the other hand, mCoCDaR results (plotted below) show that the time series of mCoCDaR risk contributions exhibit quite different patterns compared to CoVaR and CoCVaR, and compared across different institutions. With multiple regression, marginal risk contributions of each institution change significantly over time.

**Table 1.** Systemic Risk Contribution Ranking Summary.

|      | mCoCDaR | CoCDaR | mCoCVaR | CoCVaR |
|------|---------|--------|---------|--------|
| JPM  | 9       | 8      | 4       | 4      |
| BAC  | 1       | 2      | 2       | 5      |
| C    | 7       | 10     | 10      | 10     |
| WFC  | 10      | 1      | 7       | 1      |
| STT  | 4       | 9      | 6       | 6      |
| PNC  | 8       | 7      | 5       | 3      |
| USB  | 3       | 5      | 9       | 9      |
| COF  | 5       | 4      | 8       | 7      |
| BK   | 6       | 6      | 3       | 8      |
| BTT  | 2       | 3      | 1       | 2      |

Furthermore, we plot time dependent drawdowns and mCoCDaR contributions; see Figures 2–11. Each institution graph on the left plots its drawdown curve in blue versus the orange curve showing drawdowns of the Dow Jones index in the same time period, both based on cumulative uncompounded returns on a weekly basis. Every graph on the right plots fraction contribution to the total systemic risk from an individual bank. This fraction is obtained by normalizing individual contributions measured by $\Delta mCoCDaR$ described in Section 3.5. Normalization is done by dividing individual contributions by the total contribution from the ten banks. By construction, the normalized contributions sum up to one for each time step. As a result of applying the mCoCDaR regression setting, we observe that individual contributions significantly vary over time as well as across institutions. Moreover, risk contributions may have different signs. For instance, JPM and WFC always have negative contributions (see, Figures 2 and 5). Citigroup starts with negative contributions and moves to contributing positively (see, Figure 4), while the others always have positive contributions.

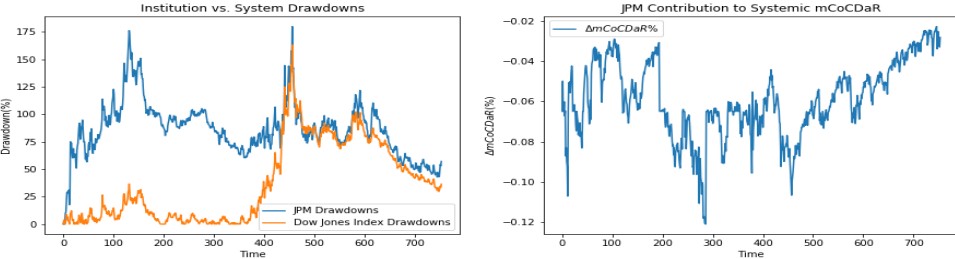

**Figure 2.** JP Morgan Chase & Company.

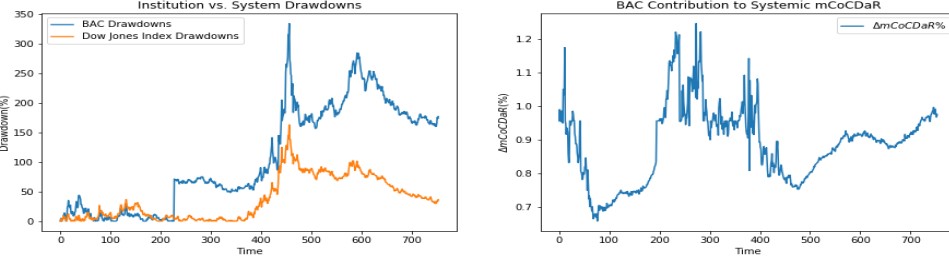

**Figure 3.** Bank of America.

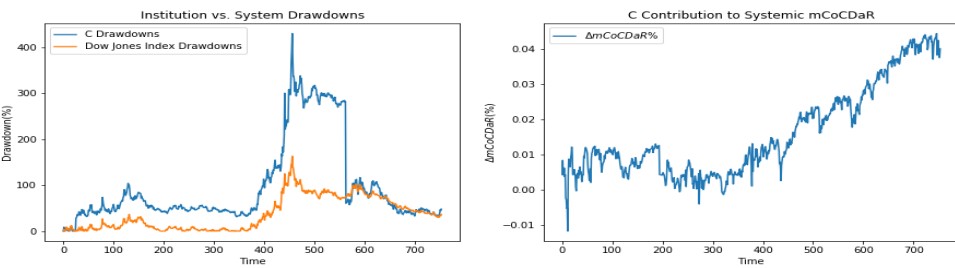

**Figure 4.** Citigroup Inc.

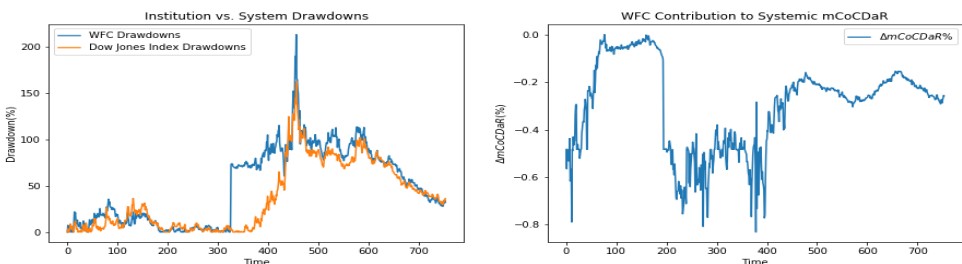

**Figure 5.** Wells Fargo & Company.

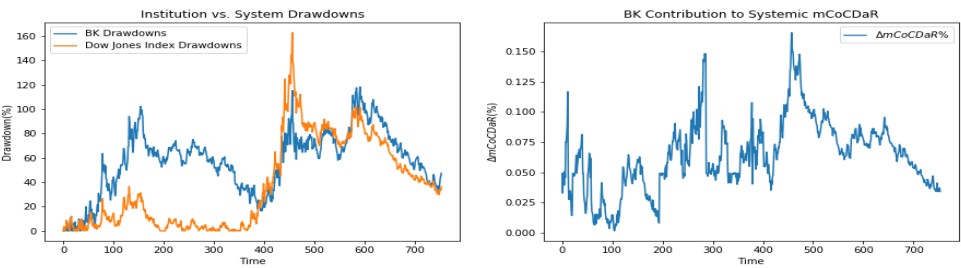

**Figure 6.** The Bank of New York Mellon Corporation.

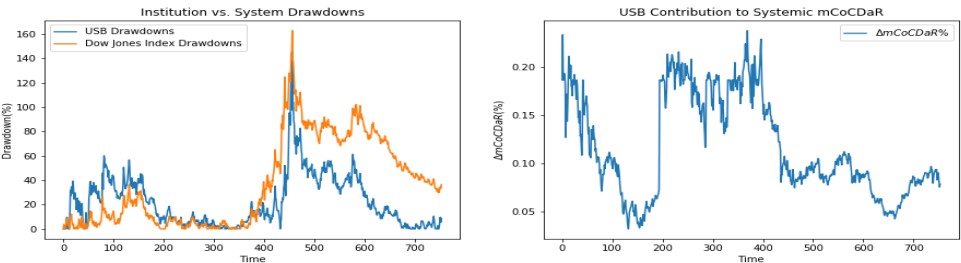

**Figure 7.** US Bancorp.

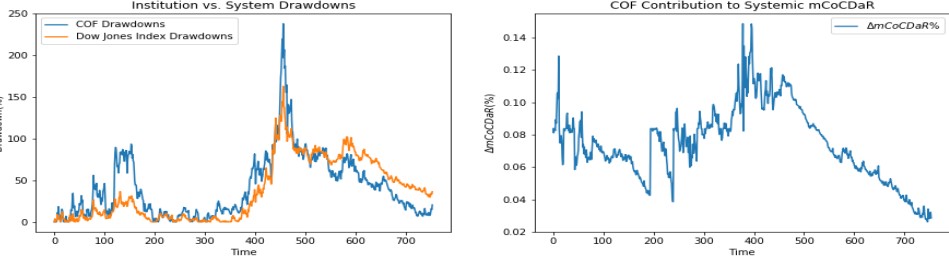

**Figure 8.** Capital One Financial Corporation.

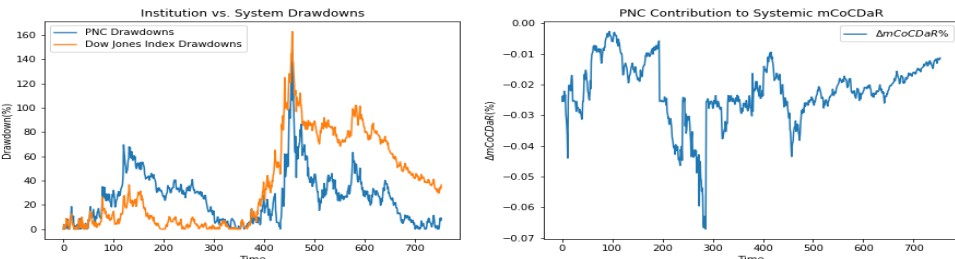

**Figure 9.** PNC Financial Services Group Inc.

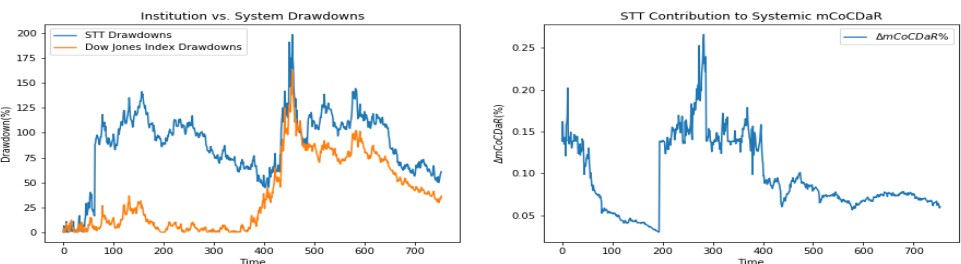

**Figure 10.** State Street Corporation.

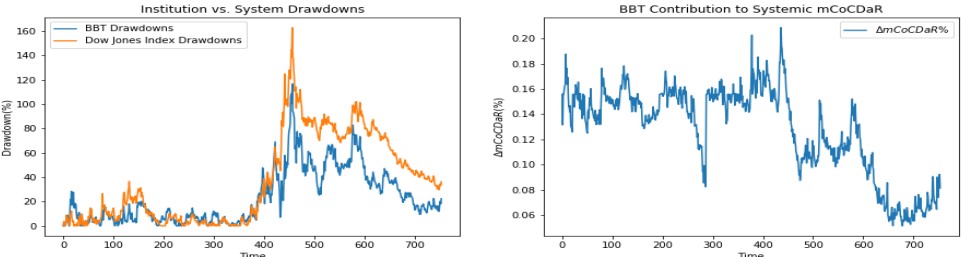

**Figure 11.** The BB&T Corporation.

## 4. mCoCDaR Application to Style Classification

This section extends the approach to hedge fund style classification. We show how to estimate CDaR as a function of drawdowns of several market indices. Style classification is a well studied topic approached by Sharpe (1992) and Carhart (1997) with a standard regression (for returns of instruments). Furthermore, it was extended by Bassett and Chen (2001) using quantile regression. Here, we demonstrate results with the mCoCDaR method. This classification explains fund drawdowns, as a function of drawdowns of several market indices (as factors). Codes, data, and results for this case study are posted at this link[4].

Similar to Bassett and Chen (2001), we investigated dependence of drawdowns of the Magellan fund (fund) from four indices: Russell 1000 value index (rlv), Russell 1000 growth index (rlg), Russell 2000 value index(ruj), and Russell 2000 growth index (ruo). These indices correspond to four equity classes: large value stocks, large growth stocks, small value stocks, and small growth stocks.

---

4　http://uryasev.ams.stonybrook.edu/index.php/research/testproblems/financial_engineering/case-study-style-classification-with-mcocdar-regression/

　　　We used a dataset from a quantile regression style classification case study posted at[5]. Golodnikov et al. (2019) considered the same dataset for testing CVaR regression, which is posted at this link[6]. The dataset contains 1264 weekly return observations for the fund and indices.

　　　We calculated drawdowns for the fund and the four indices using weekly returns in the considered time period. CVaR regression of drawdowns is done by minimizing CVaR2 error in PSG[3], as follows,

$$D_{fund,t} = \beta_0 + \beta_1 D_{rlv,t} + \beta_2 D_{rlg,t} + \beta_3 D_{ruj,t} + \beta_4 D_{ruo,t} ,$$

where $D_{i,t}$ for $i = rlv$, $rlg$, $ruj$, $ruo$ are (uncompounded) drawdowns of index $i$ at time $t$ and $D_{fund,t}$ are (uncompounded) drawdowns of the fund. See definition of drawdowns of a financial instrument in Section 2.1.

　　　For 0.9-CVaR regression the pseudo-R square equals 0.91 and the estimated coefficients are:

$$\hat{\beta}_0 = 0.3713, \ \hat{\beta}_1 = 0.4621, \ \hat{\beta}_2 = 0.5493, \ \hat{\beta}_3 = -0.0171, \ \hat{\beta}_4 = -0.0591 .$$

　　　We considered also 0.0-CVaR regression, which estimates mean and corresponds to an ordinary least squares regression. Pseudo-R square equals 0.91 and estimated coefficients are:

$$\hat{\beta}_0 = -0.2733, \ \hat{\beta}_1 = 0.4891, \ \hat{\beta}_2 = 0.5150, \ \hat{\beta}_3 = -0.0618, \ \hat{\beta}_4 = -0.0003 .$$

　　　Regression coefficients show that both large and average drawdowns of the Magellan fund are mostly explained by drawdowns in large value stocks index (coefficient $\hat{\beta}_1$) and large growth stocks index (coefficient $\hat{\beta}_2$). The fund exhibits roughly 50–50% mix of these two classes of stocks in the sense of drawdown behavior.

　　　Furthermore, we compared these results with previous studies, which used CoVaR- and CoCVaR-based measures. The CoVaR approach based on quantile regression[5] (see Problem 1 in the link) gives the following coefficient estimate:

$$\hat{\beta}_0 = -0.0089, \ \hat{\beta}_1 = 0.4602, \ \hat{\beta}_2 = 0.5176, \ \hat{\beta}_3 = -0.0156, \ \hat{\beta}_4 = 0.0001 .$$

and the CoCVaR approach based on CVaR regression[6] (see Problem 1, $\alpha$=0.9 in the link) gives the following estimate:

$$\hat{\beta}_0 = 0.0105, \ \hat{\beta}_1 = 0.6058, \ \hat{\beta}_2 = 0.4721, \ \hat{\beta}_3 = -0.0778, \ \hat{\beta}_4 = -0.0071 .$$

　　　We observe that this particular dataset considered regressions of a similar style with around a 50–50% mix of two stock indices.

## 5. On Portfolio Optimization with mCoCDaR and mCoCVaR

　　　Previous sections defined and tested mCoCDaR and mCoCVaR multiple regression versions for systemic risk measurement. It should be considered that risk measures can be used for other purposes. For instance, we can build a portfolio minimizing CoCVaR or CoCDaR, conditioned on the distress level of several market indices (or factors), under the constraint that the expected return meets some target. Similar problems were studied in Kurosaki and Kim (2013a, 2013b) with CoAVaR and CoVaR measures for conditional risk. Here, we present portfolio optimization problems using mCoCVaR and mCoCDaR risk measures:

$$\min_{\vec{w}_t} \ \ mCoCVaR_{\alpha,t}^{\vec{w}_t|f_t^1,...,f_t^K} \ \ s.t. \ \ \sum_{i=1}^{I} w_t^i r_t^i = r^\star, \ \ \sum_{i=1}^{I} w_t^i = 1$$

$$\min_{\vec{w}_t} \quad mCoCDaR_{\alpha,t}^{\vec{w}_t | f_t^1,...,f_t^K} \quad s.t. \quad \sum_{i=1}^{I} w_t^i r_t^i = r^\star, \quad \sum_{i=1}^{I} w_t^i = 1$$

where $K$ is the number of market index factors, $f_t^1,...,f_t^K$ are risk levels at time $t$ of $K$ factors (market indices), $\vec{w}_t$ is vector of portfolio weights for $I$ stocks, $r_t^i$ is return of stock $i$ at time $t$, and $r^\star$ is a target return. Systemic risk-driven portfolio selection problems were also studied in Capponi and Rubtsov (2019), where they considered portfolio optimization given a systemic event. Detailed analysis of these portfolio optimization problems is beyond the scope of this paper. We have included a short description to show that considered risk measures can be used in various areas of finance.

## 6. Conclusions

This paper proposed a new systemic risk measure, CoCDaR, which is based on conditional drawdown-at-risk and inspired by the CoCVaR approach from Huang and Uryasev (2017). We further extended the approach to mCoCDaR, which calculates conditional drawdown-at-risk of the financial system conditioned on all the institutions' drawdown distress levels. These measures can rank institutions according to their incremental (marginal) contributions to the systemic risk of the system, conditional on other institutions' distress levels. The multiple regression setting is applied to the CoCVaR measure from Huang and Uryasev (2017) and resulted in so-called mCoCVaR. Since mCoCDaR and mCoCVaR are based on multiple regression, they have the flexibility to measure joint contributions of multiple institutions. These measures are also well-defined Shapley value functions with desirable mathematical properties for a risk contribution measure. After normalization, individual risk contributions sum up to one. These advantages do not come at any additional computational cost.

CoCDaR and mCoCDaR measures are based on drawdowns (path dependent cumulative losses). These two measures capture the impact of an institution's drawdowns on the financial system's drawdowns, which is particularly suitable for market crash situations. They are useful for determining which institution may lead to a bigger crash in the market in terms of large drawdown events.

We performed a case study for the three proposed methods, CoCDaR, mCoCVaR, and mCoCDaR, using data from the ten largest banks and the Dow Jones Index, along with some state factors. The case study with codes and data are posted on the web. We have also reproduced the case study for CoCVaR measure from Huang and Uryasev (2017), with corrected signs in the returns data. We compared the ranking of institutions based on contributions to system's CoCDaR, mCoCDaR, mCoCVaR, and CoCVaR. The difference in applying CVaR- and CDaR-based measures is observed from quite different rankings of institutions. Multiple regression identifies key drivers in systemic risk because effects are marginalized. We compared time dependent curves of risk contributions for mCoCDaR and CoCVaR. Risk contributions based on CoCVaR are quite similar across institutions, while those based on mCoCDaR have very different patterns. These different patterns are implied by both the use of drawdowns and the use of multiple regressions.

Other applications of the proposed method include fund style classifications based on mCoCDaR regression. We have conducted a case study analyzing drawdowns of the Magellan fund as a function of drawdowns of four market indices. We have posted this case study to the web. The suggested methodology may also be used in various other areas of finance. In particular, we have stated portfolio selection problems with mCoCVaR or mCoCDaR objectives and constraints on expected returns.

**Author Contributions:** R.D. and S.U. stated the problem; S.U. obtained the data and provided the software; R.D. processed the data, wrote the programs and obtained the results; R.D. and S.U. analyzed the results; R.D. prepared the first draft manuscript; R.D. and S.U. revised the text and the conclusions. All authors have read and agreed to the published version of the manuscript.

**Funding:** This research received no external funding.

**Conflicts of Interest:** The authors declare no conflict of interest.

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
