# Peer review of "CoCDaR and mCoCDaR: New Approach for Measurement of Systemic Risk Contributions"

_jrfm, doi:10.3390/jrfm13110270_

Round 1

Reviewer 1 Report

This is a very interesting paper and requires the following minor modifications:

  1. I redefine the abstract of the paper in order to represent better your research for the readers because there are many acronyms.
  2. I introduce better math formulas or “translate” them for unexpert people to understand your process in the methodology.
  3. I think in the article there are few references.
  4. In my opinion, it is very excellent work. I accept the paper after minor revision.

Author Response

  1. Redefine the abstract of the paper in order to represent better your research for the readers because there are many acronyms.                  

        Response:

        We added explanation of the acronyms in the abstract for CoVaR (Value-at-Risk conditioned on an institution), CoCVaR (Conditional Value-at-Risk conditioned on an institution), and CoCDaR (Conditional Drawdown-at-Risk conditioned on an institution).   

  1. Introduce better math formulas or “translate” them for unexpert people to understand your process in the methodology.                              

        Response:

        We explained each regression problem in words and referred to papers discussing those regression methods in detail, particularly those relevant to quantile regression and CVaR regression.

  1. I think in the article there are few references.   

        Response:

        We added references in the main body of the paper. 

Reviewer 2 Report

This is an interesting and well written paper introducing and studying a new systemic risk measure called CoCDaR, financial system (index) conditional drawdown at risk driven by a drawdown level of a particular institution (equity) or a set of institutions. The methodology presentation is quite clear and I have only a few minor remarks to the case study:

  • The presentation of the CoCDaR results on p. 10 should be better. The large number of decimals is not needed, two or three would be sufficient. The numbers should be displayed in a table also with the CoCVaR results that are not reported. The same applies to the results reported on p. 11-12.
  • I assume that the units are %, but is should be clearly stated. I assume that the estimates are as of Jan 30, 2015, i.e., based on the full period. Fig. 2-11 indicate that the values are calculated dynamically over the period. What is the data time step, one day?
  • The CocVaR should be defined technically in the paper. Is it one day CVaR conditional on an institution’s CVaR? Why are the reported numbers on p.11 negative when CVaR is by definition positive and larger value indicates larger risk?
  • Figures 2-11 indicate that the CoCDaR measure is quite unstable overtime. Is not it a disadvantage of the proposed measure? The sharp jump between day 400 and 500 of the observation window in case of many banks looks strange and should be explained.

Author Response

  1. The presentation of the CoCDaR results on p. 10 should be better. The large number of decimals is not needed, two or three would be sufficient. The numbers should be displayed in a table also with the CoCVaR results that are not reported. The same applies to the results reported on p. 11-12. Response:                                                                                              We corrected some numerical results and changed the reporting format. The CoCVaR results are added to Section 3.2. All results are reported using 5 decimals. We summarized the ranking results in Section 3.6. The numeric values of the contributions have different meanings for different measures so we didn't put them in one single table for comparison.
  1. I assume that the units are %, but it should be clearly stated. I assume that the estimates are as of Jan 30, 2015, i.e., based on the full period. Fig. 2-11 indicate that the values are calculated dynamically over the period. What is the data time step, one day?                                                              Response:                                                                                            The reported estimates are averages over the full time period and the plot shows the entire time series week by week. The reported values are now converted to units of returns so there are no percentage signs needed. The plots are plotted with percentage scales and we have labelled them clearly.
  1. The CoCVaR should be defined technically in the paper. Is it one day CVaR conditional on an institution’s CVaR? Why are the reported numbers on p.11 negative when CVaR is by definition positive and larger value indicates larger risk?                                                                                                  Response:                                                                                              CoCVaR is the CVaR conditioned on an institution's returns, we referred to Huang and Uryasev for details of this measure. We changed the sign to positive after correcting numerical results.
  1. Figures 2-11 indicate that the CoCDaR measure is quite unstable overtime. Is not it a disadvantage of the proposed measure? The sharp jump between day 400 and 500 of the observation window in case of many banks looks strange and should be explained.                                                      Response:                                                                                          After correcting a sign error in the index data, we obtained new results that are more stable and the above jump in contributions doesn't occur. However, we expected some large changes in risk contributions because drawdown based regressions give more drastic changes than if returns are used. We have used a multiple regression which accounts for the dependencies between banks.

Reviewer 3 Report

A file with the report is attached.

Author Response

  1. It should be CDaR(X) = CVaR(Y)                                                        Response:                                                                                                We have changed the order in this equality.
  1. The mathematical definition (a formula) of DaR should be included. Probably, it could be placed in Section 2.1.                                                    Response:                                                                                             We have added details of DaR in section 2.5 where it first occurred. It is the VaR    of drawdowns. We added a formula for VaR.

  1. I suggest to set α'1= 0.9 and α'2= 0.5. The question is that the difference should depend on the choice of example.                                        Response:                                                                                             We have changed the notation in section 2.6 and 2.9. Since we made it by default that 0.9 represents a stress level and 0.5 is by default the normal level, we didn't include further subscripts on the notation for readability purposes.

  1. I suggest to delete indexes 1,...,I from α' since α' is the same for all i = 1,...,I. And to change it in the way of remark 3.                                    Response:                                                                                            The indices are used mainly to differentiate each institution and suggest that you can calculate more complicated contributions based on different combinations of joint contributions, although in our case we only calculated differences between 0.9 and 0.5 for one institution as an example. The rest of the institutions need to be fixed at 0.5 level so we use this notation to distinguish which institution is the one being considered in terms of risk contributions.